# Norepinephrine Inhibits Lipopolysaccharide-Stimulated TNF-α but Not Oxylipin Induction in n-3/n-6 PUFA-Enriched Cultures of Circumventricular Organs

**DOI:** 10.3390/ijms23158745

**Published:** 2022-08-06

**Authors:** Fabian Johannes Pflieger, Jacqueline Wolf, Martin Feldotto, Andreas Nockher, Tatjana Wenderoth, Jessica Hernandez, Joachim Roth, Daniela Ott, Christoph Rummel

**Affiliations:** 1Institute of Veterinary Physiology and Biochemistry, Justus Liebig University Giessen, 35392 Giessen, Germany; 2Institute for Laboratory Medicine, Pathobiochemistry, and Molecular Diagnostics, University Hospital of Giessen and Marburg, 35043 Marburg, Germany; 3Center for Mind, Brain and Behavior (CMBB), University of Marburg and Justus Liebig University Giessen, 35032 Marburg, Germany

**Keywords:** cytokines, lipopolysaccharide, oxylipins, circumventricular organs, immune-to-brain communication

## Abstract

Sensory circumventricular organs (sCVOs) are pivotal brain structures involved in immune-to-brain communication with a leaky blood–brain barrier that detect circulating mediators such as lipopolysaccharide (LPS). Here, we aimed to investigate the potential of sCVOs to produce n-3 and n-6 oxylipins after LPS-stimulation. Moreover, we investigated if norepinephrine (NE) co-treatment can alter cytokine- and oxylipin-release. Thus, we stimulated rat primary neuroglial sCVO cultures under n-3- or n-6-enriched conditions with LPS or saline combined with NE or vehicle. Supernatants were assessed for cytokines by bioassays and oxylipins by HPLC-MS/MS. Expression of signaling pathways and enzymes were analyzed by RT-PCR. Tumor necrosis factor (TNF)α bioactivity and signaling, IL-10 expression, and cyclooxygenase (COX)2 were increased, epoxide hydroxylase (Ephx)2 was reduced, and lipoxygenase 15-(LOX) was not changed by LPS stimulation. Moreover, LPS induced increased levels of several n-6-derived oxylipins, including the COX-2 metabolite 15d-prostaglandin-J2 or the Ephx2 metabolite 14,15-DHET. For n-3-derived oxylipins, some were down- and some were upregulated, including 15-LOX-derived neuroprotectin D1 and 18-HEPE, known for their anti-inflammatory potential. While the LPS-induced increase in TNFα levels was significantly reduced by NE, oxylipins were not significantly altered by NE or changes in TNFα levels. In conclusion, LPS-induced oxylipins may play an important functional role in sCVOs for immune-to-brain communication.

## 1. Introduction

Supplementation of long-chain polyunsaturated fatty acids (PUFA) enriched for omega-3 PUFA (n-3) have been shown to exhibit beneficial modulatory effects in a variety of diseases, including aging [1], Parkinson’s and Alzheimer’s disease, depression, or anxiety in rodent models [2] and human clinical studies [3,4]. Oxylipins compose a large group of PUFA-derived oxidation-derived metabolites involving a variety of enzymes, including cyclooxygenase (COX)2, lipoxygenases (LOX), P450 enzymes, and soluble epoxide hydroxylase (sEphx2) [5,6,7]. These metabolites have been implicated in various processes in health and disease such as pain, oxidative stress, and inflammation [8]. In particular, an important role for conveying pro- and anti-inflammatory effects from n-6 versus n-3-derived oxylipins has been proposed [9,10]. Moreover, oxylipin profiles are novel biomarkers for detection of disease states [8,11] and cellular phenotype polarization such as in A1-like and A2-like astrocytes induced by lipopolysaccharide (LPS) or cytokine treatment [12]. Indeed, oxylipin production and their functional significance have been studied in vitro, for example, in macrophages [13], microglia [14,15], astrocytes [12,16,17], neurons [18], and primary human brain microvessel endothelial cells [19]. However, other than studies on hippocampal abundance [20,21], brain-region-specific production has rarely been investigated for oxylipins.

Interestingly, systemic, i.e., peripheral inflammation such as abdominal inflammation or lung inflammation has emerged as an important contributor for the exacerbation of preexisting brain pathology such as neurodegeneration or traumatic insults [22,23,24]. Experimentally, effects of peripheral insults on the brain have been extensively studied using intraperitoneal (i.p.) stimulation with the bacterial mimetic LPS [25,26]. Mechanisms of immune-to-brain communication during systemic/peripheral inflammation that contribute to the exacerbation of brain pathology include immune cell trafficking and humoral mediators such as the proinflammatory cytokines interleukin (IL)-6 and tumor necrosis factor (TNF)α [27]. In addition, quick neuronal signaling via sensory or vagal afferents has been described to convey inflammatory information to the brain [27,28,29]. Detection of circulating inflammatory mediators from the periphery occurs in brain endothelial cells and via sensory circumventricular organs (sCVOs), brain structures with a leaky blood–brain barrier and pivotal sites for pathways in immune-to-brain communication [30]. Our and other groups have previously revealed spatiotemporal genomic activation via nuclear factor (NF)κB or signal transducer and activator of transcription (STAT)3-signaling in sCVOs during various peripheral inflammatory insults involving direct action of LPS and IL-6 on brain cells such as astrocytes, microglia, endothelial cells, and neurons [27,31].

Additionally, during stressful events, norepinephrine (NE) is released and can act as a humoral mediator on all organ systems including sCVO brain cells. Previous reports confirmed the presence of adrenergic receptors and the functional significance of NE signaling in various brain regions [32], including sCVOs [33,34,35]. As such, in vitro stimulation with NE dampened LPS-induced inflammatory responses such as production of TNFα in microglia [36,37,38], while COX-2 expression was shown to be increased [39].

Here, we aimed to investigate whether sCVOs show changes in oxylipin release during LPS-induced inflammation. In addition, we wanted to test if NE alters cytokine release, inflammatory signaling, oxylipin release, and expression of enzymes involved in their production.

For this purpose, we used pooled primary neuroglial cell cultures of the three sCVOs, namely the vascular organ of the lamina terminalis (OVLT), subfornical organ (SFO), and area postrema (AP). Moreover, the effect of NE on the inflammatory response, i.e., TNFα and IL-6 release and inflammatory NFκB- and STAT3-signaling, was assessed via bioassays and expression of their negative regulatory proteins inhibitor (I)κBα and suppressor of cytokine signaling (SOCS)3, respectively [27]. As several oxylipins are known to convey their effects via peroxisome proliferator-activated receptor (PPAR)γ activation [40], we used (PPAR)γ coactivator 1-alpha (PGC1α) mRNA expression as a marker for its activation [41]. IL-10 mRNA expression served as an indicator for potential anti-inflammatory action in sCVOs [42]. In addition, expression levels of enzymes involved in oxylipin production were investigated (i.e., COX-2, ALOX15, and Ephx2).

## 2. Results

### 2.1. Analyses of Cytokines in Supernatants: LPS Stimulated IL-6 and TNFα Release Is Partially Inhibited by NE (TNFα) in Primary Neuroglial sCVO-Cultures

Increased TNFα and IL-6 release from cultured sCVO cells was quantified in supernatants indicative for its inflammatory potential, as previously shown [43]. Pro-inflammatory LPS-stimulation (4 h duration) significantly increased the release of TNFα (*p* < 0.0001; all *p* values represent the main effect of LPS stimulation if not stated differently) and IL-6 (*p* < 0.0001) into the supernatants (Figure 1). Additional treatment of cells with NE as a humoral signal related to psychological stress significantly dampened LPS-induced TNFα (post hoc LPS+vehicle vs. LPS+NE *p* < 0.01, Figure 1a). However, IL-6 levels were not significantly altered by NE in comparison to solely LPS-stimulated cell cultures (Figure 1b). Moreover, NE treatment also affected non LPS-stimulated cells to lower basal TNFα and IL-6 release (main effect NE treatment TNFα *p* = 0.0077, IL-6 *p* = 0.0431, Figure 1a,b).

### 2.2. mRNA-Expression Analyses of Inflammatory Marker Proteins: LPS-Induced Inflammation Increased IL-6 and TNFα-Activated Signaling Pathways, the Anti-Inflammatory Cytokine IL-10, and the Lipid-Metabolizing Enzymes COX-2, but Decreased Ephx2

mRNA expression levels of pro- and anti-inflammatory markers and enzymes regulating lipid mediator synthesis were used to further assess lipid metabolism and cytokine signaling pathways. Inhibitors of (I)κBα and SOCS3, respectively, were used as common indicators and negative feedback inhibitors of TNFα and IL-6 signaling via the proinflammatory transcription factors NFκB and STAT3 [44,45]. IκB- and SOCS3-mRNA expression showed a robust significant increase in LPS-stimulated groups (Figure 2a,b) confirming a proinflammatory activation of cells via the NFκB- and STAT3 pathway (IκB *p* < 0.0001, SOCS3 *p* = 0.0383). NE treatment only tended to lower IκB mRNA expression, suggesting that the contribution of TNFα to this response was limited (Figure 2a).

As a read-out for anti-inflammatory PPARγ pathway activation, a target of several lipid mediators such as docosahexaenoic acid (DHA) [46], neuroprotectin D1 (NPD1) [47], 15-deoxy-prostaglandin-J2 (15d-PGJ2) [48], or 13-hydroxyoctadecadienoic acid (13-HODE) [49], we measured PGC1α mRNA expression, which was not significantly affected by LPS stimulation or NE treatment (Figure 2c). As TNFα and the anti-inflammatory cytokine IL-10 are strongly interconnected [31,38,39,43,44,45,46,47,48,49,50,51,52,53,54,55,56,57,58,59], we also analyzed IL-10 mRNA expression (Figure 2d). The mRNA expression highly increased in LPS-stimulated groups (*p* = 0.0019); however, NE treatment did not significantly influence IL-10 expression after LPS stimulation.

Lipid mediator production from PUFA increases during inflammation and is related to stimulated changes of mRNA expression of involved pathway enzymes [6]. Here, COX-2 mRNA expression (Figure 2e) significantly increased in LPS-stimulated cells compared to control groups (*p* = 0.0022), whereas Ephx2 mRNA expression (Figure 2g) was significantly lower in LPS- vs. saline-stimulated counterparts (*p* = 0.0084). NE treatment did not influence COX-2- nor Ephx2-mRNA levels. Arachidonate 15-lipoxygenase (ALOX15) mRNA expression did not significantly change between groups (Figure 2f).

### 2.3. LC MS/MS Analyses of n-6 Oxylipins: Almost All n-6 Metabolites Derived from Arachidonic Acid (AA) and Linoleic Acid (LA) That Were Altered by LPS Stimulation Were Elevated

Having observed changes in LPS-induced mRNA expression and cytokine release of the sCVOs that have direct access to circulating LPS and NE during septic and psychological stress, our next aim was to determine if LPS and altered levels of TNFα by NE may affect the production of n-6-derived AA and LA metabolites. Conditioned supernatants obtained from primary sCVO cell cultures stimulated with LPS or saline and treated with NE or vehicle were analyzed by liquid chromatography–tandem mass spectrometry (LC-MS/MS) for the presence and quantity of lipid mediators and metabolites from PUFA. Lipid mediators produced from n-6 AA substrate include important proinflammatory mediators such as prostaglandin (PG)E2, which was not significantly altered under the present experimental condition and timing. However, the 15-LOX-derived AA metabolites 15-hydroxy-eicosatetraenoic acid (HETE, *p* = 0.0444); 8,15-DiHETE (*p* < 0.0001) and 15-oxo-eicosatetraenoic acid (OxoETE, *p* = 0.0002) significantly increased in supernatants of LPS-stimulated cells (Figure 3a–c). 5-LOX derived 5-OxoETE (Figure 3d) also showed significantly higher concentrations in LPS-stimulated groups compared to saline-stimulated controls (*p* = 0.009). Cytochrome P450 monoxygenase (CYP450) enzymes metabolize AA into 18-HETE, which was significantly reduced in LPS-stimulated samples (*p* < 0.0001, Figure 3e). Other CYP450 and Ephx2 metabolites such as 11,12-dihydroxy-eicosatrienoic acid (DHET, *p* = 0.0161) and 14,15-DHET (*p* < 0.0001) increased due to LPS stimulation (Figure 3f,g). Additionally, the concentration of 15d-PGJ2, an anti-inflammatory mediator that is metabolized from PGJ2 [60], was significantly higher in LPS-stimulated groups (*p* = 0.0264, Figure 3h).

n-6 LA is a precursor of AA with a slightly different composition, namely a shorter fatty acid chain. Interestingly, anti-inflammatory LA metabolite 13-hydroxy-octadecadienoic acid (HODE, Figure 3i) synthesized via 15-LOX activity significantly increased in supernatants from LPS-stimulated cells (*p* = 0.0002). Similarly, CYP450 derived 9,10-dihydroxy-octadecenoic acid (DiHOME, *p* = 0.0471) and 12,13-DiHOME (*p* < 0.0001) were increased in supernatants by LPS stimulation (Figure 3j,k). Other AA or LA metabolites detected did not significantly change with LPS stimulation or NE treatment. Overall, NE did not significantly change the concentration/fold change of any AA or LA derived metabolites, suggesting that TNFα and NE do not play a major role in LPS-induced lipid metabolism.

### 2.4. LC MS/MS Analyses of n-3 Oxylipins: n-3 Metabolites Derived from DHA Namely NPD1 Increased While EPA-Derived Metabolites Decreased or Increased after LPS Stimulation

The metabolites deriving from n-3 eicosapentaenoic acid (EPA) have mostly anti-inflammatory capacities [9,10,61]. Lipoxin A5 (LXA5) is produced via 5-LOX and 15-LOX and was significantly increased by LPS stimulation vs. saline counterparts (*p* < 0.0001, Figure 4a), whereas 12-LOX- and CYP450-derived 12-hydroxy-eicosapentaenoic acid (HEPE, Figure 4d) were reduced due to LPS-stimulation (*p* < 0.0001). CYP450 and LOX activity are also involved in the production of 8-HEPE and 9-HEPE (Figure 4b,c). Both were significantly reduced in LPS-stimulated samples compared to saline controls (8-HEPE *p* = 0.0021, 9-HEPE *p* < 0.0001). However, for CYP450-derived 18-HEPE we observed significantly higher levels in supernatants from LPS-stimulated cells (*p* = 0.0286, Figure 4e). Notably, 18-HEPE is the precursor of the anti-inflammatory and pro-resolving resolvin E1, suggesting increased production of resolvin E1 in the present study. While we were not able to detect resolvin E1 in this study, 18-HEPE is a suitable and more easily detectable mediator and anti-inflammatory marker, as previously shown [6,9,10,62].

Moreover, NPD1 (Figure 4f), an important anti-inflammatory mediator synthesized from DHA via 15-LOX [47], was strongly increased in supernatants of LPS-stimulated cells (*p* < 0.0001). Overall, other EPA or DHA metabolites detected did not significantly change with LPS stimulation and NE did not alter any measured n-3 fatty acid metabolites.

## 3. Discussion

In the present study, we revealed that LPS mainly increases n-6-derived oxylipins, while n-3 oxylipins were increased or decreased by the inflammatory stimulus in primary neuroglial cultures of rat sCVOs. This response was accompanied by LPS-induced increased NFκB and STAT3 signaling, COX-2 expression, and reduced Ephx2 mRNA expression, as well as an increased expression of the anti-inflammatory cytokine IL-10. Moreover, we show that NE inhibits LPS-induced TNFα release into the supernatants of sCVO cultures whereas oxylipin levels in supernatants and LPS-induced IL-6 levels were not altered by NE.

sCVOs have direct access to circulating mediators during systemic inflammatory insults. Indeed, we have previously shown nuclear translocation of STAT3 and NFκB and calcium responses to cytokines such as IL-6 and TNFα and pathogen-associated molecular patterns such as LPS in these brain structures using in vitro and in vivo approaches [27,31,43,51,63]. In addition, lipid mediators such as PGE2 induce increased intracellular calcium concentration alone or enhance glutamate-induced responses in neurons and astrocytes of the OVLT involved in the fever induction pathways [52]. Interestingly, PGE2 can also reduce LPS-induced release of TNFα in these cultures, most likely representing a negative feedback loop to limit the inflammatory response [52]. Here, we expand on these findings and establish, for the first time, an altered production of several n-6 and n-3 derived oxylipins in these pivotal brain structures that are essential for immune-to-brain communication. To ensure the availability of substrates and to mimic in vivo supplementation, we enriched the cultures with AA, EPA, and DHA (50 µmol/L) during cultivation, as described previously [55]. Supplementation of the cultures was relevant, as the sCVOs would have direct access to such PUFA through the circulation. While such enrichment can be regarded as a limitation of the current study, we aimed to investigate the capacities of sCVO brain cells to convert such substrates, which necessitates their high abundance.

NE treatment decreased LPS-induced levels of bioactive TNFα, but not IL-6, in supernatants of primary sCVO cultures. This result is partially in line with previous reports showing that NE, at the same dose as applied here, reduced TNFα release from primary rat and mouse microglia cultures [36]. However, we did not observe a significant reduction in IL-6 bioactivity, at least after LPS stimulation. This difference may be related to a rather high variance between independent cell-culture experiments, the brain structures used for extraction (cortex versus sCVOs), and the fact that our cultures contained rather high amounts of astrocytes (~50%), oligodendrocytes (~30%), followed by around ~10% of neurons and ~10% microglia. Interestingly, the oxylipin 18 HEPE, a precursor of resolvin E1, has been shown to reduce TNFα sec secretion from murine macrophages at 0.5 µM concentration [64]. As 18-HEPE was elevated by LPS in supernatants in the present study, this mechanism may not have been involved but may have masked some more robust effects by NE.

LPS stimulation was also accompanied by further inflammatory signaling as evidenced by increased mRNA expression of IκBα and SOCS3, negative regulatory proteins and markers of NFκB or STAT3 activation, as previously reported in sCVOs [27,65,66]. Interestingly, while oxylipins have been reported to act via PPAR activation [40], we did not reveal any significant changes in PGC1α mRNA expression. In addition to proinflammatory cytokines, the anti-inflammatory cytokine IL-10 was also significantly increased by LPS but not altered by NE. Regarding important enzymes involved in oxylipin production, COX-2-expression was increased by LPS but its expression was not further enhanced by NE as previously reported [39]. Here, we applied a much higher LPS dose than applied by Schlachetzki et al. (2010; 10 µg/mL versus 10 ng/mL), which occurs during septic-like inflammation in mice [67]. This may have resulted in a ceiling effect, which was not further affected by NE. Nonetheless, we believe that it was meaningful to test if NE, as a mimetic of a systemic stress response, can modulate LPS-induced oxylipin production. Future studies should include further pharmacological testing of epinephrine, NE, and acetylcholine as indicators to the action of both the sympathetic and the parasympathetic nervous system, on oxylipin production in sCVOs.

In fact, our results provide support for the notion that cytokines such as TNFα are not major contributors to the observed changes in oxylipin production, as changes in its bioactivity did not lead to changes in oxylipin levels. However, further studies are needed to confirm this hypothesis. While several important oxylipins (i.e., 8,15-diHETE, LXA5, and NPD1) were increased by LPS stimulation, the expression of their metabolizing enzyme ALOX15 was not altered within the sCVOs, similar to what has been described in the hippocampus after in vivo LPS stimulation [21]. Interestingly, ALOX15 expression was increased 24 h after LPS stimulation (1 µg/mL) in a murine microglial cell line in vitro; suggesting potential cell-type specific changes that may not have been picked up in our culture with only 10% microglial contribution [68]. sEphx2 expression was analyzed, as this enzyme is increased in depressive patients and brains of animal models of depression-like behavior, and its inhibition is a promising target for future treatment strategies of depression [69]. Additionally, it has been proposed that the sEphx2 enzyme could be involved in the metabolic inactivation of oxylipin mediators [5]. Here, we observed its reduced expression during LPS-induced inflammation in sCVOs, suggesting potential reduced activation.

Despite the fact that experimental animal work has revealed exciting new data on beneficial direct and indirect effects of n-3 PUFA on a variety of brain diseases and insults [70], data on n-3 PUFA and brain inflammation and neurodegeneration in humans remains controversial [71]. Mixed results most likely pertain to differences in the study design, dose, and timing of PUFA supplementation, as well as the notion that n-3 PUFA may not always be the treatment of choice, e.g., when host inflammatory responses are altered during some kinds of infections [72,73,74]. Data for the situation in humans are largely lacking [75]; nonetheless, recent evidence indicates that some mechanisms proposed in animal models may also play a role in humans. Using a human hippocampal progenitor cell line, Borsini et al. (2021) showed that LOX and P450 EPA/DHA metabolites such as 18-HEPE, which was also detected in the present study, may contribute to the prevention of apoptosis and increase neurogenesis. Importantly, these oxylipins were also found in plasma of patients following treatment with their precursors and correlating with less severe depressive symptoms [76].

Concerning n-6-derived oxylipins, several candidates were increased by LPS-stimulation, including 15-HETE, 13-HODE, and 15d-PGJ2 [77], as well as 5-oxo-ETE and 15-oxo-ETE [78], known to act as PPARγ ligands with partially potent anti-inflammatory capacities [40]. As such, LPS has been shown to induce 15d-PGJ2 in primary rat microglial cultures [79], and Huang et al. (2015) revealed inhibition of microglial activation by i.p. 15d-PGJ2 treatment in a model of brain ischemia [80], potentially via NFκB inhibition [81]. Moreover, 15d-PGJ2, which is also found in cerebrospinal fluid, inhibited fever during systemic LPS-induced inflammation when intracerebroventricularly infused [60]. In particular, larger amounts of 15-HETE can be detected in supernatants of IL-4-induced M2-polarized macrophages after live *E. coli* [82] or LPS stimulation (100 ng/mL, 16 h) [13]. Oxidized eicosanoids such as 5-oxo-ETE and 15-oxo-ETE were also increased by LPS-stimulation in the present study. As such, it has been demonstrated that 5-oxo-ETE production increases during inflammation [83] and may play a role in the pathophysiology of allergic inflammation to induce skin infiltration of eosinophils by calcium mobilization [84]. In respect to the functional significance of increased n-6-derived oxylipins described here, similar mechanisms could also apply for sCVOs. Therefore, such mechanisms may be related to immune-cell trafficking to the sCVOs, polarization of sCVO microglia in a macrophage-like manner, or anti-inflammatory action of oxylipins (e.g., 15d-PGJ2) to counteract and dampen systemic inflammatory immune-to-brain signaling. The precise role of each of these mediators within the sCVOs remains to be further investigated in future studies.

Concerning n-3-derived oxylipins, we found three major entities of mediators—EPA-derived LXA5, 18-HEPE, and DHA-derived NPD1—to be increased, and several HEPE species (8-HEPE, 9-HEPE, 12-HEPE) to be decreased by LPS in the supernatants of sCVO cultures. Interestingly, while not much is known about the functional significance of LXA5, it was shown to be involved in superoxide anion generation in canine neutrophil granulocytes [85], which may be also relevant for the brain during septic-like inflammation. More studies have already investigated the role of 18-HEPE, the precursor of resolvin E1, for its anti-inflammatory pro-resolving effects in peripheries such as the lung [62,86] and the brain [68,69]. For example, 18-HEPE has been demonstrated to induce brain-derived neurotrophic factor in Müller glia [87] or to inhibit TNFα secretion after LPS stimulation (0.5 µg/mL) in a murine macrophage cell line [64]. Moreover, 18-HEPE is also found in supernatants of M1 and M2 macrophages [13]. In addition, NPD1 remains one of our most prominently detected mediators increased by LPS in our sCVOs cultures. This small pro-resolving lipid mediator (SPM) has multiple beneficial effects during brain inflammatory insults such as ischemic or traumatic brain injury [88] via preservation of mitochondrial membrane structure [89] or blockage of neuronal apoptosis and antioxidative effects [90]. However, caution needs to be taken as this mediator is hard to distinguish from its isomer protectin DX [91]. Nonetheless, PDX has also been shown to exhibit less prominent but similar beneficial anti-inflammatory effects [92] such as inhibition of LPS-induced COX-2 and reactive oxygen species in human neutrophil granulocytes [93].

A recent discussion has emerged on detectability and exact enzymatic pathways for SPM production in biological samples [94]. Unfortunately, we were not able to detect SPMs such as resolvins in the supernatants of our sCVOs cultures. Nonetheless, compelling evidence supports an important function of SPMs [9], which may reach physiologically relevant concentrations in a paracrine manner, asking for further improvements and harmonization in methods for their extraction and detection. Future directions for the detection of brain oxylipins include in vivo detection by solid-phase microextraction [95]. Moreover, oxylipins represent very interesting biomarkers for disease states [8,11,16,96] or changes in cellular phenotypes between more pro- or anti-inflammatory states, e.g., in astrocytes, monocytes, and macrophages [12,13,97], with high potential as future biomarkers for stratification of patients, for example, with major depressive disorders [98]. Currently, the investigation into oxylipins and their role in health and disease is a growing field of research that could have broad implications. Our present results add new insights to existing literature for sCVOs during acute inflammation. In the last decade, modern analytical methods vastly improved the understanding of oxylipins’ role and enabled the detection of fragile metabolites as well as new interactions with pathway enzymes. A broad overview of the relationship between enzymes, oxylipins, and their substrates is shown in Figure 5, contextualizing our results. However, such an approach can only capture a simplified high-level scheme of a continuously changing body of evidence for underlying pathways [6,99,100].

Aside from technical challenges in differentiating NPD1 and PDX, the detectability of metabolic pathways, and SPMs, limitations of the present study pertain to our measures of mRNA expression of enzymes. It is well-known that mRNA levels do not necessarily reflect changes on protein level. Moreover, we did not account for dynamic changes of inflammatory signaling and oxylipin production. However, previous work has already shown partially similar but also distinct changes in enzyme expression. These experiments included assessments performed at 2 h and 24 h time points in vitro (microglia cell line) or in vivo (LPS-treatment in mice) of different brain structures such as the hippocampus [21,68]. Thus, while future studies should further expand on our present findings, such as through assessing metabolic pathways in a more dynamic manner and testing additional stimulants than LPS, we believe that our present data are meaningful and highlight the pivotal role of sCVOs and the need to explore the functional significance of oxylipin signaling in specific brain regions.

In summary, we revealed that LPS increased inflammatory signaling and altered enzyme expression, which was accompanied by increased or decreased levels of n-3- and n-6-derived oxylipin subsets. While NE inhibited TNFα release, neither n-3 nor n-6 oxylipins were significantly altered. In conclusion, we present the first evidence for the production of oxylipins in sCVO, pivotal brain structures in immune-to-brain communication. Some of these lipid mediators are already known for their capacity to modulate inflammatory responses, highlighting their potential functional significance in inflammatory signaling in the brain. In particular, NPD1 and 18-HEPE or LXA5 represent interesting targets for further functional characterization in these brain structures known to detect circulating mediators. While NE is able to modulate inflammatory TNFα production in the brain, neither NE nor TNFα seem to play a major role for local LPS-induced oxylipin release from sCVO brain cells. Overall, we highlight metabolic pathways in oxylipin production (Figure 5) and the importance of sCVOs to produce lipid mediators linked to inflammation. Locally produced oxylipins may play an important role in immune-to-brain communication within sCVOs during LPS-induced inflammation.

## 4. Materials and Methods

### 4.1. Animals

All experiments were performed in accordance with the German Animal Welfare Act and international legislation (Directive 2010/63, European Community) and were approved by local authorities (JLU number 580_M). Wistar rat pups (4–6 days old) of both sexes obtained from an in-house breeding colony were used for all experiments. Breeding animals were originally obtained from Charles River WIGA (Sulzfeld, Germany). Adult rats had ad libitum access to drinking water and standard laboratory chow; dams reared their pups in M4-size cages. Room temperature was controlled at 22 ± 1 °C, relative humidity at 50%, and 12 h light cycles.

### 4.2. Isolation and Cultivation of OVLT, SFO, and AP Primary Cell Cultures

Best results for successful preparation of differentiated rat brain neuroglial cell cultures can be achieved from neonatal rat pups. Under these conditions, neurons and glial cells respond to a variety of transmitters and stimuli typical for the respective brain structure, such as glutamate or LPS [31,43,50,51,52,53]. Primary cell cultures of all sCVOs were established from topographically excised brain tissue of 4–6-day-old Wistar rat pups. Six animals per preparation (balanced male and female pups) were quickly decapitated with sharp scissors and the heads were immersed (<20 s) in cold 70% ethanol. Each brain was immediately removed from the skull, under aseptic conditions, and fixed onto a Teflon^®^-block with Histoacryl^®^ tissue glue (Braun, Melsungen, Germany). The brains were rapidly transferred to a chamber containing ice-cold oxygenated Gey’s Balanced Salt Solution (GBSS; Biotrend, Cologne, Germany) enriched with 5% d-glucose (Sigma-Aldrich, Munich, Germany) where serial coronal brain slices (400 μm) were cut starting at the level of the anterior hypothalamus using a vibratome (752 M Vibroslice; WPI, Berlin, Germany). Sections containing the OVLT were selected with the anterior commissure and optic chiasm as main neuroanatomical landmarks (bregma ∼0.20 mm–∼−0.26 mm) [54]. SFO sections were collected based on the dorsal opening of the third ventricle, the SFO extending into the ventricular space and its vicinity to ventricle’s choroid plexus (bregma ∼−0.80 mm–∼−0.92 mm). The AP (bregma ∼−13.68 mm–∼−13.8 mm) was located on the dorsal surface of the medulla immediately adjacent to the *Nucleus tractus solitarius* (NTS). OVLT, SFO, and AP regions were dissected under stereomicroscopic control (SMZ-U; Nikon, Düsseldorf, Germany) using fine eye scissors. For OVLT and SFO sections, it should be noted that 400 μm coronal brain slices used represent more than just one coronal plane from the representative stereotaxic atlas of the rat brain [54]. Therefore, borders at the rostral and caudal end of prepared tissue slices were identified by apparent neuroanatomical structures.

The isolated tissue from OVLT, SFO, and AP was transferred into Petri dishes containing oxygenated HBSS (Hanks Balanced Salt Solution without Ca^2+^ and Mg^2+^; Biochrom, Berlin, Germany) supplemented with 20 mM HEPES (Sigma Aldrich) pH 7.4. The supernatant was removed and tissue samples from all three regions were pooled and treated with 2 mL dispase (2 U/mL; Roche Diagnostics, Mannheim, Germany) in oxygenated HBSS with 20 mM HEPES, pH 7.4, at 37 °C. After 45 min of incubation, the tissue was washed three times with 1 mM EDTA (Merck KGaA, Darmstadt, Gemany) in HBSS to inactivate the enzyme. Subsequently, EDTA was sluiced down from tissue with 3 mL complete medium consisting of Neurobasal medium A supplemented with 2% B 27, penicillin (100 U/mL)/streptomycin (0.1 mg/mL), and 2 mM l-glutamine (all Life Technologies GmbH, Darmstadt, Germany). Sedimentation of tissue fragments was followed by removal of supernatants and tissue was resuspended in 1 mL complete medium. Repeated trituration with a 1 mL Eppendorf pipette was applied to dissociate tissue fragments. The dissociated cells were counted using a hemocytometer, diluted in complete medium to 120,000 cells/mL and plated onto prewarmed, poly-l-lysine (1 mg/mL H_2_O; Sigma Aldrich)-coated CELLocate^®^ glass coverslips, which were mounted on reusable Flexiperm-micro-12 wells (6 mm diameter; Heraeus, Hanau, Germany). After that, cells were cultured in a humidified atmosphere of 5% CO_2_/95% air at 37 °C. Preliminary experiments applying immunohistochemical detection of cellular marker proteins revealed a mean cellular content as follows: astrocytes (~50%), oligodendrocytes (~30%), ~10% of neurons, and ~10% microglia.

### 4.3. Experimental Protocol

After seeding, cells were incubated for 24 h to adhere to the coated glass cover slips. Medium was replaced to remove debris and cultures were incubated for an additional 48 h. On the fourth day of cultivation, medium was replaced for 24 h with fresh medium supplemented with AA, EPA, and DHA, each 50 µmol/L, as described previously [55]. On the fifth day, fatty-acid-enriched medium was removed and cultures washed with normal medium and stimulated with 10 µg/mL LPS (*Escherichia coli* O111:B4, Sigma-Aldrich Chemie GmbH), at a dose that induces strong inflammatory activation [31,43] or saline. Moreover, groups received an additional treatment with 1 µmol/L NE (norepinephrine-hydrogen-tartrate, Sigma Aldrich) or vehicle according to a previously published protocol [39]. All reagents were diluted in medium. After 4 h, experiments were stopped by collecting supernatants and lysing cells from coverslips by washing with saline and applying 200 μL lysis buffer for 2 min (RA1buffer, NucleoSpin© RNA XS kit, Macherey Nagel, Düren, Germany). Cells and supernatants were stored at −80 °C until further analysis. Each sample consisted of 6 culture wells with a pooled volume of 2.1 mL supernatant coming from 252,000 seeded cells on coverslips. Sample size was *n* = 6 of pooled samples per group with four treatment groups investigated in all six independent preparations. Additional spare cell-suspension wells were incorporated into the cultivation process and stained for vitality (Trypan blue solution, Sigma Aldrich) at the end of experiments to assure cultures’ quality.

### 4.4. Cytokine Measurements

sCVOs cell cultures’ supernatants were analyzed for IL-6 levels by a bioassay based on the dose-dependent growth-inducing effect of IL-6 on a B9 hybridoma cell line. Concentrations of TNFα in supernatants were quantified using the cytotoxic effect of TNFα on a WEHI cell line as described previously by Welsch et al. (2012) [56]. For calibration of the bioassays, we used international standards (murine TNFα standard: code 88/532; human IL-6 standard: code 89/548; National Institute for Biological Standards and Control, South Mimms, UK). After adjusting for the dilution of the samples, the bioassays’ detection limits were determined to be 3 IU IL-6/mL and 6 pg TNFα/mL, respectively.

### 4.5. Eicosanoid Extraction and LC-MS/MS Based Mass Spectrometric Analysis

All LC-MS-grade solvents were purchased from VWR, Merck (both Darmstadt, Germany), Honeywell (Seelze, Germany), and Fisher Scientific (Schwerte, Germany). Eicosanoid standards and deuterated internal standards were obtained from Cayman Chemical (local distributor: Biomol, Hamburg, Germany).

Eicosanoids were extracted and analyzed by high-pressure liquid chromatography–tandem mass spectrometry (HPLC-MS/MS), as described previously [57,58]. In brief, 1 mL of cell-culture supernatants was mixed with 25 µL of the deuterated internal standard mixture and 50 µL methanol and vortexed vigorously. Samples were centrifuged (10,000 rpm, 4 °C, 5 min) and the supernatants were subjected to solid-phase extraction via Bond Elute Plexa solid-phase extraction columns (Agilent Technologies, Santa Clara, CA, USA). Extraction was performed following the manufacturer’s instructions and eicosanoids were eluted using 500 µL methanol. The solvent was evaporated and extracts were resolved in 100 µL of water/acetonitrile/formic acid (70:30:0.02, *v*/*v*/*v*; solvent A) and immediately analyzed by HPLC-MS/MS.

Eicosanoids were analyzed on a QTRAP 5500 mass spectrometer (Sciex, Darmstadt, Germany), coupled to an Agilent 1290 Infinity LC system (Agilent Technologies, Santa Clara, CA, USA), equipped with a Synergi Hydro reverse-phase C18 column (2.1 × 250 mm; Phenomenex, Aschaffenburg, Germany). The flow rate was set to 0.3 mL/min and analytes were separated applying the following gradient: 1 min (0% solvent B: acetonitrile/isopropyl alcohol, 50:50, *v*/*v*;), 3 min (25% solvent B), 11 min (45% solvent B), 13 min (60% solvent B), 18 min (75% solvent B), 18.5 min (90% solvent B), 20 min (90% solvent B), 21 min (0% solvent B). The column was re-equilibrated by keeping 0% solvent B for five min. Eicosanoids were analyzed in negative-ion mode applying a scheduled multiple-reaction monitoring approach, using transitions, individual collision energies, and further instrument parameters, as described by Dumlao et al. [57]. For quantification, an external 10-point calibration curve was measured in triplicate. Data analysis and peak integration were performed using the MultiQuant software (v.2.1.1, Sciex, Darmstadt, Germany). The table of results can be found in Appendix A.

### 4.6. Real Time RT-PCR

For determination of gene expression of selected target genes involved in inflammation and lipid mediator pathways, we performed real-time RT-PCR. RNA was extracted from harvested cell samples using the RNA isolation kit (NucleoSpin^®^ Macherey-Nagel, Düren, Germany) according to the manufacturer’s protocol. Briefly, cell lysates were homogenized and filtered to remove debris. RNA and DNA were bound by Ethanol and caught in a column filter. Next, DNA was digested by rDNase mixture and rinsed from the column. Finally, pure RNA was eluted from the column and collected in single reaction tubes. Extracted RNA was stored at −20 °C for later reverse transcription. About 200 ng total RNA was employed for reverse transcription by 50 U of murine leukemia virus reverse transcriptase, 40 μM random hexamers, and 10 μM deoxynucleoside triphosphate (dNTP) mix (Applied Biosystems, Foster City, CA, USA), which were added to a total reaction volume of 20 μL. The StepOnePlus Real-Time PCR System (Applied Biosystems) was used for relative quantification of all samples in duplicates applying a primer/probe mixture (TaqMan Gene Expression Assay, Applied Biosystems) and a TaqMan PCR Master Mix (Applied Biosystems) with the following cycling protocol: polymerase activation (50 °C for 2 min), initial denaturation (95 °C for 10 min), 40 cycles of denaturation (95 °C for 15 s), and annealing/elongation (60 °C for 1 min). We applied the following gene-expression assays from Applied Biosystems: ALOX15 (Rn01646191_m1), COX-2 (Rn01483828_m1), Ephx2 (Rn00576023_m1), GPC1α (Rn00580241_m1), IL-10 (Rn99999012_m1), IκB (Rn01473657_g1), and SOCS3 (Rn00585674_s1). For analysis of cDNA quantities between samples and groups, we beforehand normalized the individual gene expression to the housekeeping gene β-actin (Rn00667869_m1; Applied Biosystems) as reference in each sample, which exhibited stable expression between treatment groups. For relative quantification of every gene’s expression, the 2^−(ΔΔCt)^ method was calculated so that results represent the x-fold difference in relation to a control sample, which showed the lowest expression.

### 4.7. Data Analysis and Statistics

Relative quantity values were chosen for direct statistical analysis of gene mRNA expression. Parameters measured from supernatants (cytokines, lipid mediators) were normalized before statistical analyses to fold change of the LPS+vehicle-treated group within each independent experiment to account for differences caused by environmental factors between days of cell-culture preparation. Cytokines and lipid mediators were quantified by regression analysis of standard curve functions. For lipid mediators detected by LC-MS/MS, cell-culture medium was analyzed as control samples. In case of a detectable lipid mediator showing smaller quantities in the sample than the mean concentration in medium controls, these sample values were set to the mean concentration of medium controls, which were used as a lower cut off values. Supernatant lipid mediator concentrations were LOG10-transformed before normalization to the LPS+vehicle group. In general, groups were analyzed by two-way ANOVA with the factors of stimulation (LPS, saline) and treatment (NE, vehicle). Main effects were reported and in case of significant interaction, Bonferroni post hoc testing was performed for the single-group comparison. *p* < 0.05 was applied for statistical significance and mean ± SEM are displayed in the figures.

## Figures and Tables

**Figure 1 ijms-23-08745-f001:**
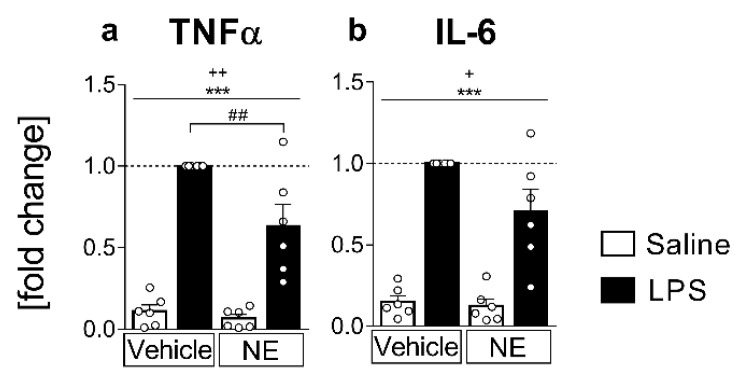
Bioassay cytokine measurements from supernatants: Release of the proinflammatory cytokines tumor necrosis factor α (TNFα, (**a**) and interleukin (IL)-6 (**b**) into the supernatants of primary sensory circumventricular organ (sCVO) cultures. Cells were stimulated with lipopolysaccharide (LPS, 10 µg/mL) or saline and simultaneously treated with norepinephrine (NE, 1 µmol/L) or vehicle for 4 h. For every sample set, cytokine concentrations are plotted as fold change to the LPS-treated sample (LPS+vehicle) from six independent experiments. TNFα and IL-6 secretion increased after LPS stimulation (* main effect LPS stimulation), whereas NE treatment lowered basal and LPS-induced secretion of both cytokines into the supernatants (+ main effect NE treatment). LPS-induced TNFα release was significantly dampened by NE (# LPS+vehicle vs. LPS+NE). *n* = 6 per group; two-factorial ANOVA, Bonferroni post hoc test, *** *p* < 0.001; ^+^
*p* < 0.05, ^++^
*p* < 0.01; ## < 0.01.

**Figure 2 ijms-23-08745-f002:**
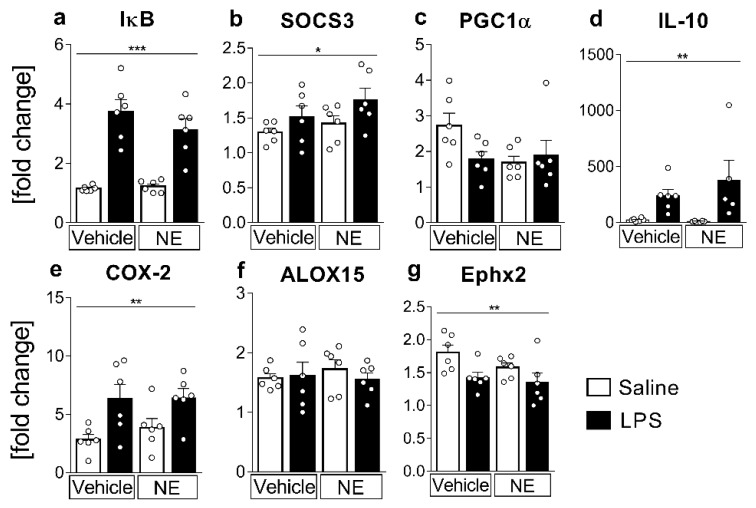
mRNA expression of inflammatory markers and lipid mediator metabolism: mRNA expression was assessed for markers of inflammatory pathway activation (**a**–**d**) and lipid mediator metabolism enzymes (**e**–**g**). Cells were stimulated with lipopolysaccharide (LPS, 10 µg/mL) or saline and simultaneously treated with norepinephrine (NE, 1 µmol/L) or vehicle for 4 h. The 2-ΔΔct method was applied to present relative quantities as fold change of the lowest expression. Pro-inflammatory pathway-activation-marker inhibitor (I)κB, (**a**) and suppressor of cytokine signaling (SOCS)3, (**b**) and the anti-inflammatory cytokine interleukin (IL)-10 (**d**) increased due to LPS stimulation. The rate-limiting enzyme for prostaglandin production, i.e., cyclooxygenase (COX)-2 (**e**), showed an LPS-induced increase, whereas there were reduced epoxide hydrolase (Ephx)2 (**g**) mRNA levels in LPS-stimulated groups. Peroxisome proliferator-activated receptor gamma coactivator ((PGC)1α, (**c**)) and arachidonate 15-lipoxygenase (ALOX15, (**f**)) mRNA expression were not significantly affected by LPS-stimulation or NE treatment. *n* = 6 per group; two-factorial ANOVA, Bonferroni post hoc test, * main effect LPS stimulation; * *p* < 0.05, ** *p* < 0.01, *** *p* < 0.001.

**Figure 3 ijms-23-08745-f003:**
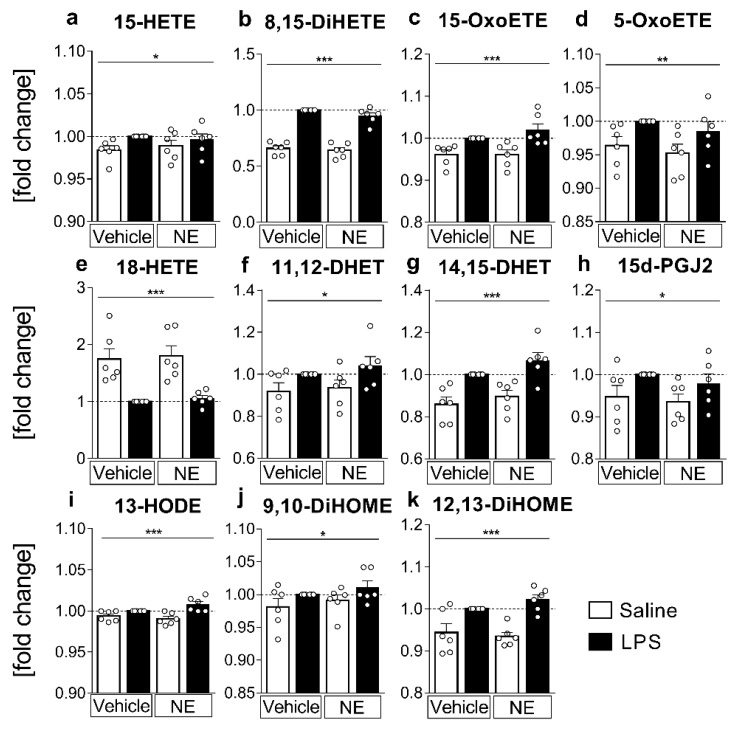
n-6 lipid mediators and metabolites from supernatants: detection and quantification of n-6 lipid mediators and metabolites derived from arachidonic acid (AA) (**a**–**h**) or linoleic acid (LA) (**i**–**k**) from supernatants using LC-MS/MS. Primary sensory circumventricular organ (sCVO) cell cultures were stimulated with lipopolysaccharide (LPS 10 µg/mL) or saline and simultaneously treated with norepinephrine (NE, 1 µmol/L) or vehicle for 4 h. For every sample set, metabolite concentrations are plotted as fold change to the LPS-stimulated sample (LPS+vehicle) for each of the six independent experiments. AA-derived metabolites elevated in LPS-stimulated samples: (**a**) 15-HETE (hydroxy-eicosatetraenoic acid); (**b**) 8,15-DiHETE (dihydroxy-eicosatetraenoic acid), (**c**) 15-OxoETE (oxo-eicosatetraenoic acid); (**d**) 5-OxoETE; (**f**) 11,12-DHET (dihydroxy-eicosatrienoic acid); (**g**) 14,15-DHET; (**h**) 15d-PGJ2 (15-Deoxy-Delta-12,14-prostaglandin-J2); (**e**) 18-HETE was reduced in LPS-stimulated samples. LPS-induced LA metabolites showed higher levels in supernatants compared to controls: (**i**) 13-HODE (hydroxy-octadecadienoic acid); (**j**) 9,10-DiHOME (dihydroxy-octadecenoic acid); (**k**) 12,13-DiHOME; *n* = 6 per group, two-factorial ANOVA, Bonferroni post hoc test, * main effect LPS stimulation, * *p* < 0.05, ** *p* < 0.01, *** *p* < 0.001.

**Figure 4 ijms-23-08745-f004:**
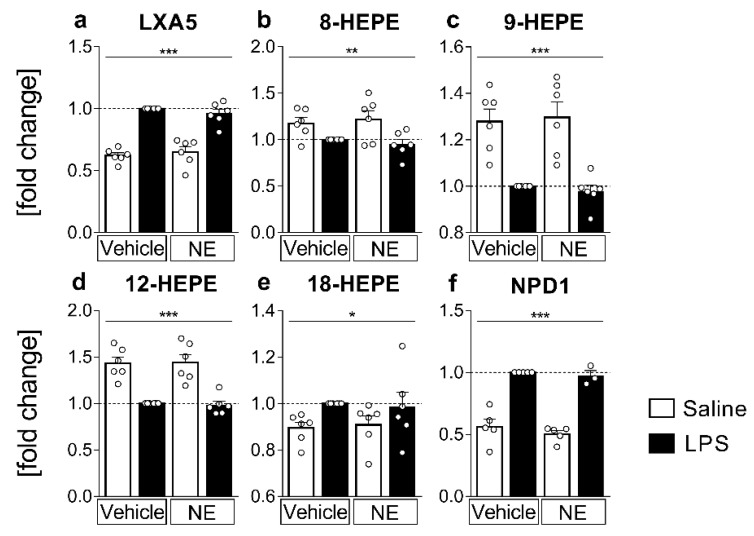
n-3 lipid mediators and metabolites from supernatants: detection and quantification of n-3 lipid mediators and metabolites derived from eicosapentaenoic acid (EPA, (**a**–**e**)) or docosahexaenoic acid (DHA, (**f**)) from supernatants using LC-MS/MS. Primary sensory circumventricular organ (sCVO) cell cultures were stimulated with lipopolysaccharide (LPS, 10 µg/mL) or saline and simultaneously treated with norepinephrine (NE, 1 µmol/L) or vehicle for 4 h. For every sample set, metabolite concentrations are plotted as fold change to the LPS-stimulated sample (LPS+vehicle) for each of the six independent experiments. The EPA metabolites LXA5 (Lipoxin A5, (**a**)) and 18-HEPE (hydroxy-eicosapentaenoic acid, (**e**)) were elevated in supernatants of LPS-stimulated cells, whereas 8-HEPE (**b**), 9-HEPE (**c**), 12-HEPE (**d**) levels were reduced due to LPS-stimulation when compared to controls. DHA-derived NPD1 (neuroprotectin D1, (**f**)) showed LPS-induced increase. *n* = 6 per group, two-factorial ANOVA, Bonferroni post hoc test, * main effect LPS stimulation; * *p* < 0.05, ** *p* < 0.01, *** *p* < 0.001.

**Figure 5 ijms-23-08745-f005:**
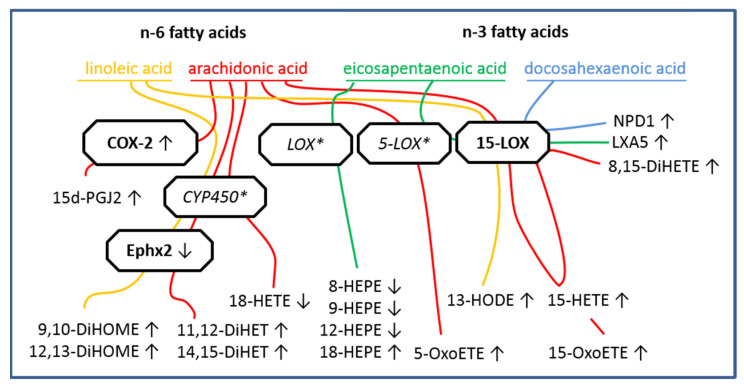
Summary of results and lipid mediator metabolism pathways in lipopolysaccharide (LPS)-stimulated CVO cultures: simplified pathway models of polyunsaturated fatty acids’ (PUFA) metabolism with literature-based “checkpoint” enzymes; n-6 fatty acids (linoleic acid, yellow; arachidonic acid, red) and n-3 fatty acids (eicosapentaenoic acid, green; docosahexaenoic acid, blue) are metabolized by the same enzymes: cyclooxygenases (COX), lipoxygenases (LOX), cytochrome P450 monoxygenases (CYP450), and epoxide hydrolase 2 (Ephx2). Other pathway enzymes are not displayed for better visualization. Increased or decreased levels of metabolites in supernatants or cells’ mRNA expression of enzymes are displayed by arrows (increased ↑, decreased ↓). Substrates (fatty acids) and enzymes not measured by us are visualized in color (fatty acids) or in italic with an asterisk (*). Colored lines depict metabolic pathways via enzymes to produce oxylipins from their substrates. Sensory circumventricular organ (sCVO)-derived primary cell cultures were pre-incubated with PUFA (each 50 µmol/L). LPS-induced (10 µg/mL) inflammatory stimulation of cell cultures for 4 h induced mRNA expression of COX-2 and decreased mRNA expression of Ephx2. ALOX15 expression was not affected by LPS stimulation. Enzymes’ activity led to altered release of lipid metabolites into supernatants, as depicted by arrows. Arachidonic acid-derived metabolites were elevated in LPS-stimulated samples: 15-HETE (hydroxy-eicosatetraenoic acid); 8,15-DiHETE (dihydroxy-eicosatetraenoic acid), 15-OxoETE (oxo-eicosatetraenoic acid); 5-OxoETE; 11,12-DHET (dihydroxy-eicosatrienoic acid); 14,15-DHET; 15d-PGJ2 (15-Deoxy-Delta-12,14-prostaglandin-J2). The arachidonic acid metabolite 18-HETE was reduced in LPS-stimulated samples when compared to controls. Linoleic acid metabolites were increased in supernatants by LPS: 13-HODE (hydroxy-octadecadienoic acid); 9,10-DiHOME (dihydroxy-octadecenoic acid); 12,13-DiHOME. Eicosapentaenoic acid metabolites LXA5 (Lipoxin A5) and 18-HEPE (hydroxy-eicosapentaenoic acid) were elevated in supernatants of LPS-stimulated cells, whereas 8-HEPE, 9-HEPE, and 12-HEPE levels were reduced due to LPS stimulation versus controls. Docosahexaenoic-derived NPD1 (neuroprotectin D1) was observed to be increased by LPS stimulation. Metabolic pathways are visualized according to [6,99,100].

## Data Availability

The data presented in this study are available in Appendix A and are available on rescannable request from the corresponding author.

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
