# Peer review of "Norepinephrine Inhibits Lipopolysaccharide-Stimulated TNF-α but Not Oxylipin Induction in n-3/n-6 PUFA-Enriched Cultures of Circumventricular Organs"

_ijms, 2022, doi:10.3390/ijms23158745_

Round 1

Reviewer 1 Report

THe work is a significant one with well designed experiments and the resuls obtained are reasonable and interesting and conclusions drawn are good and valid.

It would have bene interesting had the authors looked at the effect of epinephrine or adrenaline as well. Thus a comparison between epinephrine and nor-epinephrine would have been interesting.

Similarly a study looking at the effects of acetylcholine would have been interesting since Ach has opposite actions to epinephrine and nor-epinephrine. Such studies would have given an indication as to the actions of both sympathetic and parasympathetic nervous systems of eicosanoids and other related metabolites.

Author Response

THe work is a significant one with well designed experiments and the resuls obtained are reasonable and interesting and conclusions drawn are good and valid.

Response: We thank the referee for the positive evaluation of our manuscript.

It would have bene interesting had the authors looked at the effect of epinephrine or adrenaline as well. Thus a comparison between epinephrine and nor-epinephrine would have been interesting.

Response: We agree that such pharmacological work is of interest with the known differences in affinity of these two mediators to receptor subtypes. However, this was not the scope of the current work. To be able to analyze oxylipins, we already had to pool supernatants derived of many animals for this small brain structures.

Similarly a study looking at the effects of acetylcholine would have been interesting since Ach has opposite actions to epinephrine and nor-epinephrine. Such studies would have given an indication as to the actions of both sympathetic and parasympathetic nervous systems of eicosanoids and other related metabolites.

Response: Again, we agree that such pharmacological studies are of interest in the future and thank for this interesting suggestion but this is out of the scope of the present manuscript. To account for this suggestion. We added a comment to the revised manuscript highlighting that such studies are of future interest (lines 379-381).

“Future studies should include further pharmacological testing of epinephrine, NE and ace-tylcholine as indicators to the action of both, the sympathetic and the parasympathetic nervous system, on oxylipin production in sCVOs.”

Reviewer 2 Report

The work "Norepinephrine inhibits lipopolysaccharide-stimulated TNF-α but not oxylipin induction in n-3 / n-6 PUFA enriched cultures of circumventricular organs"  by Pflieger et al. is well written, organized and explained and of great interest. I suggest this manuscript for publication. 

Page 3 line 120: Please add a spot at the end of the sentence.

Author Response

Referee 2

The work "Norepinephrine inhibits lipopolysaccharide-stimulated TNF-α but not oxylipin induction in n-3 / n-6 PUFA enriched cultures of circumventricular organs" by Pflieger et al. is well written, organized and explained and of great interest. I suggest this manuscript for publication. 

Response: Thank you for the positive evaluation of our manuscript.

Page 3 line 120: Please add a spot at the end of the sentence.

Response: We thank you for your careful reading and have corrected accordingly.

Reviewer 3 Report

The presented study presents the results in my opinion of a very interesting research project, which I consider innovative and original in the context of the specificity of the environment in which the research was carried out. Moreover, this study is timely and scientifically sound and it is within the scope of the journal. The figures additionally facilitate the full evaluation of the content contained in the manuscript and significantly increase its value. The introduction provides an interesting admission to the topic. The methods are described in a detailed and clear manner. The discussion contains the most important information necessary to draw conclusions from the conducted research.

However, the manuscript requires editorial corrections, which will make it much easier to interpret.

The Abstract should be a brief summary of the entire manuscript. In its current form, it lacks the used methods and conclusions.

Lines from 88 to 92 should be moved to the discussion or conclusion.

There is no clear goal of the performed experiments.

For results, you need to break them down into separate sections with titles. Each of the experiments should be described in an appropriately specified section with the title. In the present form, there are no titles of the experiments and instead of them there is a conclusion from this experiment and then a description, unfortunately the whole thing is not divided into sections, which makes it difficult to find the results of individual research.

The limitations of the study should be included.

Author Response

Referee 3

The presented study presents the results in my opinion of a very interesting research project, which I consider innovative and original in the context of the specificity of the environment in which the research was carried out. Moreover, this study is timely and scientifically sound and it is within the scope of the journal. The figures additionally facilitate the full evaluation of the content contained in the manuscript and significantly increase its value. The introduction provides an interesting admission to the topic. The methods are described in a detailed and clear manner. The discussion contains the most important information necessary to draw conclusions from the conducted research.

However, the manuscript requires editorial corrections, which will make it much easier to interpret.

Response: We appreciate your interest in our study and are grateful for your constructive comments.

The Abstract should be a brief summary of the entire manuscript. In its current form, it lacks the used methods and conclusions.

Response: We have revised the abstract by highlighting the methods and adding a conclusion within the 200 word limit.

Lines from 88 to 92 should be moved to the discussion or conclusion.

Response: We have now moved the respective text to the conclusions.

There is no clear goal of the performed experiments.

Response: We apologize for not having been clear enough on our overall goal. We have added the following part to specify (line 95 to 98): “Here, we aimed to investigate whether sCVOs show changes in oxylipins release during LPS-induced inflammation. In addition, we wanted to test if NE alters cytokine release, inflammatory signaling, oxylipin release, and expression of enzymes involved in their production.”

For results, you need to break them down into separate sections with titles. Each of the experiments should be described in an appropriately specified section with the title. In the present form, there are no titles of the experiments and instead of them there is a conclusion from this experiment and then a description, unfortunately the whole thing is not divided into sections, which makes it difficult to find the results of individual research.

Response: In response to this comment, we added and formatted headlines to dissect the results into subsections to enhance clarity for the reader.

The limitations of the study should be included.

Response: We have expanded on limitations in the discussion as recommended. These pertain to our measures of mRNA-expression of enzymes, dynamics of inflammatory signalling, lipid mediator production and sensitivity of analyses (lines 486-498).

“Aside from technical challenges in differentiating NPD1 and PDX, the detectability of metabolic pathways, and SPMs, limitations of the present study pertain to our measures of mRNA expression of enzymes. It is well known that mRNA levels do not necessarily reflect changes on protein level. Moreover, we did not account for dynamic changes of inflammatory signalling and oxylipin production. However, previous work has already shown partially similar but also distinct changes in enzyme expression. These experiments included assessments performed at 2 h and 24 h time points in vitro (microglia cell line) or in vivo (LPS-treatment in mice) of different brain structures like the hippocampus [21,68]. Thus, while future studies should further expand on our present findings, such as through assessing metabolic pathways in a more dynamic manner and testing additional stimulants than LPS, we believe that our present data is meaningful and highlights the pivotal role of sCVOs and the need to explore the functional significance of oxylipin signalling in specific brain regions.”

Reviewer 4 Report

The manuscript which title is “Norepinephrine inhibits lipopolysaccharide-stimulated TNF-α but not oxylipin induction in n-3 / n-6 PUFA enriched cultures of circumventricular organs” is interesting. However, there are several questions in the manuscript. The color column in all figures should present the treatment condition. Based on the expression of mRNA could not represent the expression of protein, the authors should show the expression of protein in the figure 2. And, the authors should provide the evidence about the change of mRNA expression after LPS treatment for 4h. The authors also should revise the significantly format in figure for easy to read. There are several typo errors. Finally, the results form present study could not support their conclusion as shown in figure 5. The authors should revise the figure.

Author Response

Referee 4

The manuscript which title is “Norepinephrine inhibits lipopolysaccharide-stimulated TNF-α but not oxylipin induction in n-3 / n-6 PUFA enriched cultures of circumventricular organs” is interesting. However, there are several questions in the manuscript.

Response: Thank you for your critical review which helped to improve the revised version of our manuscript.

The color column in all figures should present the treatment condition.

Response: The graphics have been edited based on your recommendations. Now, saline- and LPS-treatment are depicted as open bars and black-filled bars, respectively for more clarity. Vehicle- and NE-treatment is labelled below the bars now.

Based on the expression of mRNA could not represent the expression of protein, the authors should show the expression of protein in the figure 2.

Response: Thank you for this important comment. Unfortunately, we are clearly limited in terms of the time (7 days for revision) to conduct additional experiments and perform analyses on protein level. Additionally, we are restricted with regard to available sample material, since only small amount of cell material can be harvested from sCVOs cultures. While we agree with the referee that such information would be interesting, we still believe that investigations of mRNA expression during acute inflammatory responses are of interest and meaningful. We now added this limitation to the discussion section to account for differences between the regulation of mRNA and protein levels as well as timing aspects. Moreover, we highlight how our results relate to previous studies that have performed similar analyses (lines 486-498).

And, the authors should provide the evidence about the change of mRNA expression after LPS treatment for 4h.

Response: Indeed, we present data of mRNA-expression 4h after LPS-treatment (see for example Figure legend): „Cells were stimulated with lipopolysaccharide (LPS, 10 µg/ml) or saline and simultaneously treated with norepinephrine (1 µmol/l) or vehicle for 4 h.” The evidence for mRNA expression is shown in Figure 2.

The authors also should revise the significantly format in figure for easy to read.

Response: We have adjusted the format to visualize LPS-treatment effects and the presentation of main effects (long line above the bars) in the figures for more clarity as suggested.

There are several typo errors.

Response: We apologize for these spelling errors. The revised manuscript has been proofread by a native English-speaking person and was adjusted accordingly.

Finally, the results form present study could not support their conclusion as shown in figure 5. The authors should revise the figure.

Response: Thank you for your assessment of the figure. Our intention of Figure 5 was not to visualize the conclusions, but to summarize all results for more clarity to the unfamiliar reader. To clarify, which enzymes were not measured, we have highlighted them now with an asterisk and adjusted the legend and coloured the substrates according to the lines representing metabolic pathways. Moreover, in response to this comment we revised the title of Figure 5 for more clarity “Summary of results and lipid mediator metabolism pathways in lipopolysaccharide (LPS)-stimulated sCVO cultures”

Reviewer 5 Report

The manuscript by Pflieger et al. on changes in oxylipins release during LPS-induced inflammation in sCVOs and showing the capability of NE to inhibit the LPS- stimulated TNF-α is well written and documented. I have only minor comments.

First in the abstract, the comment “This response [LPS-induced increase in TNFα levels was significantly reduced by NE] was accompanied by increased levels of several n-6-derived oxylipins including the COX-2 metabolite 15d-prostaglandin J2 or the Ephx2 metabolite 14,15-DHET” is not supported by the presented results.

Moreover, in Figure 5, it is required to explain the meanings of colorful lines and provide references for metabolic pathways not supported by these results.

Author Response

Referee 5

The manuscript by Pflieger et al. on changes in oxylipins release during LPS-induced inflammation in sCVOs and showing the capability of NE to inhibit the LPS- stimulated TNF-α is well written and documented. I have only minor comments.

Response: Thank you for your positive evaluation and suggestions for improvement.

First in the abstract, the comment “This response [LPS-induced increase in TNFα levels was significantly reduced by NE] was accompanied by increased levels of several n-6-derived oxylipins including the COX-2 metabolite 15d-prostaglandin J2 or the Ephx2 metabolite 14,15-DHET” is not supported by the presented results.

Response: Thank you for your attentive reading. The abstract has been adjusted accordingly.

Moreover, in Figure 5, it is required to explain the meanings of colorful lines and provide references for metabolic pathways not supported by these results.

Response: In the legend of Figure 5, we have now explained colour coding of lines and labelled substrates in the same colour as lines depicting metabolic pathways of respective substrates. In addition, we have included references to metabolic pathways that are shown in Figure 5 but which were not investigated in our study (lines 477-485 and Figure legend).

Round 2

Reviewer 4 Report

None